# TiM3: Titanium Alloy Microstructure Generation Following Multiple Mechanical Properties

## Abstract

Generating titanium alloy microstructure images according to the required mechanical properties can guide the optimization of materials and save lots of resources that would otherwise be consumed by traditional try-and-test material experiments. Existing microstructure image generation methods mainly focus on unconditional or discrete conditional applications, failing to support multiple continuous mechanical properties, such as Rockwell Hardness. Moreover, a crucial factor that determines the properties of titanium alloys, metallography, is still underexplored, which leads to the risk of unrealistic distorted structures. In this work, we propose **TiM3**, a two-stage coarse-to-fine latent diffusion model for **Ti**tanium alloy **M**icrostructure generation following **M**ultiple **M**echanical properties. We embed required mechanical properties using a probabilistic encoder to raise model robustness to unseen conditions and adopt cross-attention to guide the generation process. To enable the model to focus on the metallography of titanium alloys, we separate the generation into two stages. The first stage produces coarse metalloraphic structures as intermediate representations, and the second stage complements these with fine-grained details. To evaluate models' continuous generation capability in the high-dimensional mechanical property space, we design a property sampling algorithm to balance generalizability testing and property authenticity. TiM3 shows outstanding microstructure image quality, diversity, and property accuracy in both quantitative and qualitative experiments.

## 1 Introduction

Titanium alloys exhibit exceptional performance and are extensively used in various fields, including aerospace Zhao et al. (2022); Chakraborty et al. (2022), biomedical Sarraf et al. (2021); Han et al. (2023), marine applications Xu et al. (2021), etc. Designing titanium alloys for specific applications requires ensuring a range of mechanical properties. For instance, producing aircraft landing gear demands high-strength titanium alloys with tensile strength exceeding 1,000 MPa. Traditional design processes rely heavily on human intuition and extensive trial-and-error material experiments. However, with advancements in generative models Kingma & Welling (2013); Goodfellow et al. (2014); Van den Oord et al. (2016); Rombach et al. (2022), generating microstructure images has emerged as a promising assistance for designing titanium alloys. By analyzing the relationship between generated images and mechanical properties, we can streamline the designing process Lütjering (1999); Guo et al. (2016); Huang et al. (2022); Yi et al. (2024), significantly reducing testing costs.

Existing microstructure generation methods Iyer et al. (2019); Fokina et al. (2020); Haribabu et al. (2023); Robertson et al. (2023); White et al. (2024) predominantly focus on unconditional settings Fokina et al. (2020); Haribabu et al. (2023), discrete conditions Iyer et al. (2019); Howland et al. (2023), or simple statistics Robertson et al. (2023), which do not directly link generated images to complex and continuous mechanical property requirements, as illustrated in Figure 1a. A more effective approach, shown in Figure 1b, generates microstructure images guided by multiple required properties, which is an area that remains under-explored.

Moreover, a crucial factor that determines the mechanical properties of titanium alloys, metallography, is underexplored in the generation process. Metallography describes the composition and

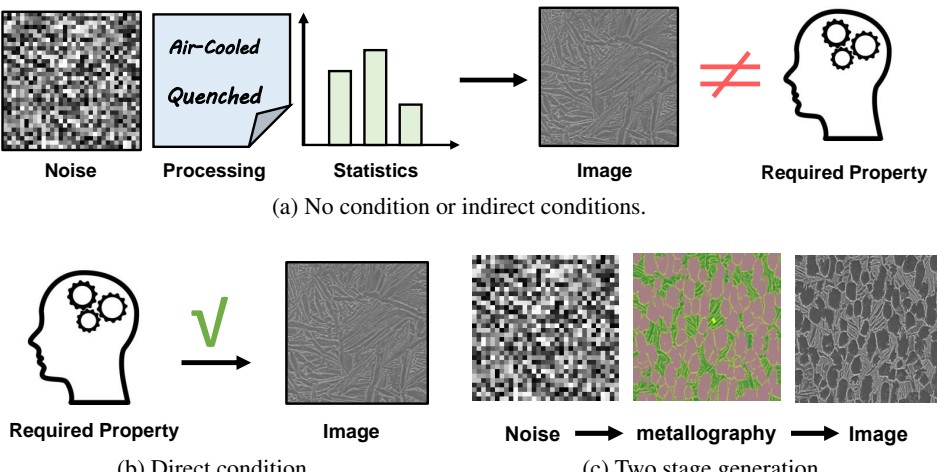

(a) No condition or indirect conditions.

(b) Direct condition.

(c) Two stage generation.

Figure 1: Motivation. (a) Existing microstructure generation methods lack direct conditioning, resulting in images that fail to meet specific property requirements. (b) Our method directly guides the generation process using the desired mechanical properties as conditions. (c) We first generate a coarse metallographic image and then generate a fine-grained realistic microstructure image.

component states of alloys and is always examined by images taken with microscopes, which intuitively show the grain morphology, phase distribution, precipitated phase size, etc. The ignorance of metallography in generation might violate the microscopic topology and distribution characteristics of real microstructures, which significantly raises the risk of structural distortion.

To address the above problems, we propose TiM3, adopting latent diffusion models to generate titanium alloy microstructure images guided by multiple mechanical properties. 1) TiM3 encodes specified properties into latent vectors and samples final embeddings from learnable Gaussian mixture models. Sampled property embeddings are then used to guide the generation process via cross-attention. Such probabilistic embedding methodology enhances the robustness to unseen continuous conditions. 2) TiM3 adopts a two-stage coarse-to-fine generation pipeline, which helps the model focus on the metallography of titanium alloys. In the first stage, TiM3 produces metallographic images as the coarse intermediate representation. In the second stage, such images are encoded and concatenated with noise latents, and the diffusion model generates realistic microstructure images, which complement fine-grained structural details.

We also developed metrics to evaluate the quality, diversity, and accuracy of the generated images to specified mechanical properties. Given the high dimensionality of the mechanical property space, exhaustively testing the entire space is impractical. To address this, we designed an efficient sampling method that uniformly samples properties within the convex hull of the collected dataset for evaluation. This approach balances the assessment of the model's generalizability and the authenticity of the properties in realistic scenarios. Based on it, we employ a pre-trained model to predict the mechanical properties of the generated images, and the accuracy is calculated accordingly.

Experiments on Ti555 alloys demonstrate the effectiveness and efficiency of TiM3. Our method achieves state-of-the-art accuracy in matching the mechanical properties while delivering outstanding results in overall quality and diversity of generated images.

Our contribution can be summarized as follows:

- We propose TiM3, generating titanium alloy microstructure images following multiple mechanical properties.

- We devise a probabilistic property encoder to enhance model robustness to unseen continuous conditions. Moreover, adopting the metallographic image as the intermediate representation, we build a two-stage coarse-to-fine microstructure generation pipeline.

- We propose a sampling method for property evaluation that effectively balances testing for generalizability and ensuring property authenticity in realistic scenarios.

- Experiments demonstrate the effectiveness of TiM3 in mechanical property accuracy, general image quality, and diversity.

## 2 RELATED WORK

### 2.1 MICROSTRUCTURE IMAGE GENERATION

Microstructure image generation has been extensively explored in computational material design Li et al. (2018); Fokina et al. (2020); Haribabu et al. (2023); Robertson et al. (2023). Li et al. Li et al. (2018) introduced an unconditional GAN-based microstructure generation pipeline, while Haribabu et. al. Haribabu et al. (2023) evaluated the performance of different models, focusing on unconditional generation. To exert greater control over the generation process, discrete conditions Iyer et al. (2019); Howland et al. (2023) have been employed. For example, Iyer et. al. Iyer et al. (2019) used five different cooling methods as conditions, while ACWGAN-GP Howland et al. (2023) conditioned on temper treatments, discretized feed rates, and tensile strength. For continuous conditions, Robertson et. al. Robertson et al. (2023) utilized 1- and 2-point statistics. However, these types of conditions do not directly address the continuous nature of material mechanical properties, which must be considered simultaneously in the generation process.

### 2.2 CONTINUOUS CONDITIONED GENERATION

Mechanical properties are continuous variables, yet traditional GAN models Mirza & Osindero (2014) and diffusion models Ho et al. (2020); Rombach et al. (2022) are typically designed for discrete conditions. CcGAN Ding et al. (2021) introduced the first generative model for images that supports continuous conditions, using vicinal risk minimization to enhance models' robustness to continuous labels and relying on kernel estimation for label smoothing. This approach has been further developed in CcDPM Zhao et al. (2024) and CCDM Ding et al. (2024). However, these methods depend on label smoothing via kernel estimation, which may be less effective in the high-dimensional space of mechanical properties, where multiple properties are specified simultaneously. This issue is exacerbated when the number of samples is limited. Alternatively, we encode mechanical properties using a continuous encoder and augment them with random noise to improve the model's robustness when training.

### 2.3 DIFFUSION MODELS

Recently, diffusion models have become prominent in the field of image generation Hertz et al. (2024); Chen et al. (2024); Hao et al. (2024). DDPM Ho et al. (2020) introduces a process where Gaussian noise is gradually added to the ground truth image, forming a Markov chain, with image generation being the inverse process. However, DDPM typically requires hundreds or thousands of steps to produce high-quality results, making it time-consuming. To improve efficiency, DDIM Song et al. (2020) modifies the inverse process, allowing for step skipping. Latent Diffusion Models (LDM) Rombach et al. (2022) further optimize this by transforming images into a smaller latent space before applying the diffusion process, significantly reducing both time and memory costs. Our work leverages LDM as the primary framework for efficient and high-quality image generation.

## 3 METHODOLOGY

### 3.1 PRELIMINARIES

**Overall Framework** The task involves generating microstructure images of titanium alloys that correspond to specified mechanical properties. The dataset consists of $N$ samples, denoted as $\mathcal{D} = \{(\boldsymbol{x}^{(i)}, \boldsymbol{y}^{(i)})\}_{i=1}^{N}$, where $\boldsymbol{y}^{(i)} \in \mathbb{R}^{K}$ represents a vector of $K$ mechanical properties, and $\boldsymbol{x}^{(i)} \in \mathbb{R}^{3 \times H \times W}$ is the corresponding microstructure image for the $i$-th sample. During training, the generative model $\mathcal{G}$ takes $\boldsymbol{y}^{(i)}$ as input and outputs the image $\hat{\boldsymbol{x}}^{(i)} = \mathcal{G}(\boldsymbol{y}^{(i)})$. For evaluation, we sample $M$ mechanical property vectors $\{\boldsymbol{c}^{(i)} \in \mathbb{R}^{K}\}_{i=1}^{M}$, and the model $\mathcal{G}$ generates images $\tilde{\boldsymbol{x}}^{(i)} = \mathcal{G}(\boldsymbol{c}^{(i)})$. To assess accuracy, we use a pre-trained ResNet-18 He et al. (2016) model $\mathcal{F} : \mathbb{R}^{3 \times H \times W} \to \mathbb{R}^{K}$ to predict corresponding mechanical properties of each generated image $\tilde{\boldsymbol{x}}^{(i)}$.

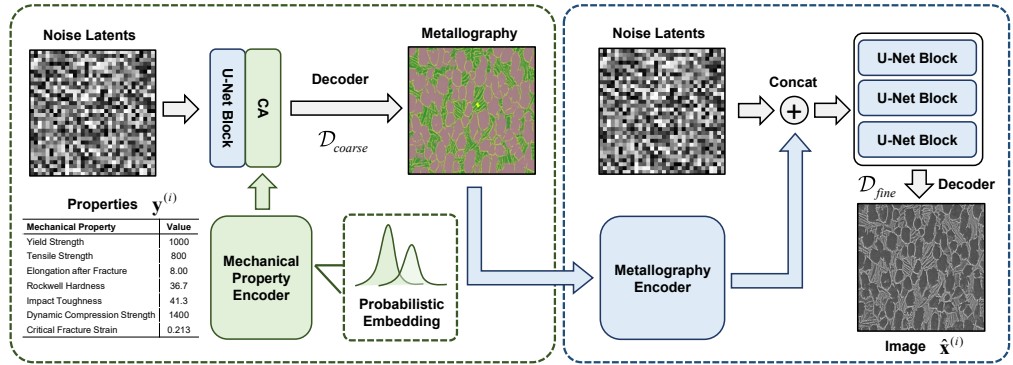

Figure 2: Overall framework. Mechanical properties are encoded into embeddings and adopted to guide the diffusion process through cross-attention. The generation is divided into two stages, where the first stage produces metallography and the second stage generates microstructure images.

The overall framework of TiM3 is illustrated in Figure 2. TiM3 adopts a two-stage generation pipeline, which first generates the metallographic image $\hat{m}^{(i)} = \mathcal{G}_1(c^{(i)})$ and then a realistic microstructure image $\hat{x}^{(i)} = \mathcal{G}_2(m^{(i)})$. Multiple mechanical properties are encoded to $\hat{g}^{(i)}$ with a probabilistic encoder, and $\hat{g}^{(i)}$ is injected into the diffusion model via vanilla cross-attention.

**Latent Diffusion Model.** We choose the latent diffusion model Rombach et al. (2022) as $\mathcal{G}$. Latent Diffusion Models (LDM)Rombach et al. (2022) have been widely used in image generation, which establishes a Markov process that gradually adds Gaussian noise to an image. LDM first encode the image $x^{(i)}$ into a latent space $z_0 = \mathcal{E}(x^{(i)}) \in \mathbb{R}^{D \times H_z \times W_z}$, where the latent shape $H_z \times W_z$ is downsampled from the input image by a factor $f$. The input latent is then perturbed as follows:

$$z_t = \sqrt{\bar{\alpha}_t} z_0 + \sqrt{1 - \bar{\alpha}_t} \epsilon_t, \epsilon_t \sim \mathcal{N}(\mathbf{0}, \mathbf{I}) \tag{1}$$

where $\bar{\alpha}_t$ controls the noise strength at each time step $t$, with $t \in 1, 2, \ldots, T$. In the reverse process, once $z_0$ is obtained, a decoder $\mathcal{D}$ is used to generate the output image: $\hat{x} = \mathcal{D}(z_0)$. During training, a random timestep $t \sim \mathcal{U}[1, T]$ is sampled, and the U-Net is trained to predict the noise at each step using the following loss function:

$$\mathcal{L} = \mathbb{E}\left[\|\epsilon - \epsilon_\theta(z_t, t, \mathcal{T}(y^{(i)}))\|\right]. \tag{2}$$

where $\mathcal{T}$ represents the encoder for the mechanical properties. During inference, we utilize DDIM Song et al. (2020), which enables denoising in fewer steps.

## 3.2 CONTINUOUS MECHANICAL PROPERTY ENCODER

To enhance the model's robustness to unseen continuous mechanical properties, we devise a probabilistic embedding method based on the Gaussian mixture model. Specifically, the mechanical property $y^{(i)}$ is first projected via a multi-layer perception $\phi_f : \mathbb{R}^K \to \mathbb{R}^D$:

$$f^{(i)} = \phi_f(y^{(i)}) \tag{3}$$

Then, the mean features and logarithmic variances of a $U$ component Gaussian mixture model are predicted via two neural networks $\phi_\mu : \mathbb{R}^D \to \mathbb{R}^{U \times D}$ and $\phi_\sigma : \mathbb{R}^D \to \mathbb{R}^{U \times D}$:

$$\mu_u^{(i)} = \phi_\mu(f^{(i)}), \quad \sigma_u^{(i)} = \phi_\sigma(f^{(i)}). \tag{4}$$

The final embedding feature follows the Gaussian mixture distribution:

$$p(g^{(i)}) = \frac{1}{U} \sum_u \mathcal{N}(\mu_u^{(i)}, \sigma_u^{(i)}). \tag{5}$$

For simplicity, we assume the variance of each Gaussian component to be diagonal. In inference, the embedding features are sampled from an approximated single-component Gaussian distribution

via the reparameterization technique. Specifically, we first sample a standard Gaussian noise $\epsilon_u \sim \mathcal{N}(0, \boldsymbol{I}_D)$ and adopt the following computation:

$$\hat{\boldsymbol{g}}^{(i)} = \frac{1}{U} \sum_u \boldsymbol{\mu}_u^{(i)} + \frac{1}{U} \sum_u \boldsymbol{\sigma}_u^{(i)} \odot \epsilon_u \tag{6}$$

### 3.3 TWO-STAGE GENERATION

In order to lower the risk of distorted structures and achieve high-quality synthesis of titanium alloy microstructures, we propose a two-stage image generation framework based on the latent diffusion model. Provided the required multiple mechanical properties, our method gradually generates microstructure images with clear structure and realistic texture.

**Stage 1: Metallographic Image Generation.** In the first stage, the latent diffusion model $\mathcal{G}_1$ takes the required multiple mechanical properties $\boldsymbol{y}^{(i)}$, and produces the semantic segmentation map of microstructure *i.e.*, the metallographic image. The ground truth metallographic image $\boldsymbol{m}^{(i)} \in \mathbb{R}^{3 \times H \times W}$ is segmented by a pretrained model, which is rendered to a pseudo RGB image. The metallographic image illustrates the primary $\alpha$ phase, $\beta$ phase, and the secondary $\alpha$ phase of the corresponding titanium alloy material sample.

**Stage 2: Microstructure Image Generation.** Based on the metallographic image $\hat{\boldsymbol{m}}^{(i)}$, the second stage latent diffusion model $\mathcal{G}_2$ complements details and produce realistic microstructure image $\hat{\boldsymbol{x}}^{(i)}$. The metallographic image is encoded via the image encoder of a frozen CLIP Radford et al. (2021) and downsampled to be concatenated with each latent noise:

$$\boldsymbol{\zeta}_{t_i} = \mathcal{F}_{\text{CLIP}}(\hat{\boldsymbol{m}}^{(i)}), \quad \boldsymbol{z}'_{t_i} = \boldsymbol{z}_{t_i} \oplus \boldsymbol{\zeta}_{t_i} \tag{7}$$

Figure 1c illustrates the metallographic image with the corresponding microstructure image. The metallographic image is a semantic map, where red regions represent $\alpha$ phase, cyan regions denote $\beta$ phase, and yellow regions indicate secondary $\alpha$ structures.

### 3.4 EVALUATION

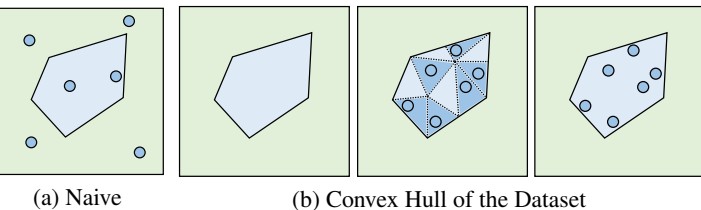

(a) Naive         (b) Convex Hull of the Dataset

Figure 3: Sampling for evaluation. The light green area represents the entire property space, while the blue polygon indicates the convex hull of the dataset.

Mechanical properties are distributed within a high-dimensional space, posing significant challenges for evaluation. In the case of 1D conditional generation, dividing the valid space into uniform buckets can be an effective way to assess model performance. However, this approach becomes impractical in high-dimensional spaces due to the curse of dimensionality and the complex distribution of properties within the dataset. A straightforward method shown in Figure 3a would be to sample $M$ property vectors from the entire space uniformly, but this often results in generating inapplicable mechanical properties that do not reflect real-world scenarios due to inherent constraints between different properties. For instance, tensile strength and impact toughness are often negatively correlated, making it unlikely to achieve high values for both properties simultaneously.

We assume the domain of each constraint to be **convex**, ensuring that property vectors within the convex hull of training dataset properties satisfy all constraints. Therefore, we sample $M$ property vectors uniformly within the convex hull. The process of this method is shown in Figure 3b and detailed in Algorithm 1. Specifically, we partition the convex hull into simplices and select $M$ simplices uniformly, weighted by their volumes. We then uniformly sample one property vector

---

**Algorithm 1** Mechanical Property Sampling

---

**Input**: $N$ vectors of mechanical properties of the dataset $\{\boldsymbol{y}^{(i)} \in \mathbb{R}^K\}_{i=1}^N$
**Parameter**: The number of sampled properties $M$
**Output**: Sample properties $\{\hat{\boldsymbol{c}}^{(i)} \in \mathbb{R}^K\}_{i=1}^M$

1: $\boldsymbol{S} := \text{Delaunay}(\{\boldsymbol{y}^{(i)}\}_{i=1}^N)$, $\boldsymbol{S} \in \mathbb{R}^{N_D \times K}$, where $N_D$ is the number of simplices
2: $\boldsymbol{p} := \text{Sample } M$ indices from $\{1, \cdots, N_D\}$ according to volumns of each simplex $\boldsymbol{S}_i$.
3: **for** $i$ in $\{1, 2, \cdots, M\}$ **do**
4: $\quad \boldsymbol{q} := \text{Sample } K$ values from $\mathcal{U}[0, 1]$.
5: $\quad$ Sort $\boldsymbol{q}$ in ascending order.
6: $\quad \boldsymbol{q}_0 := 0, \boldsymbol{q}_{K+1} := 1$.
7: $\quad$ **for** $j$ in $\{1, 2, \cdots, K+1\}$ **do**
8: $\quad\quad \boldsymbol{w}_j := \boldsymbol{q}_j - \boldsymbol{q}_{j-1}$.
9: $\quad$ **end for**
10: $\quad \hat{\boldsymbol{c}}^{(i)} := \text{MatMul}(\boldsymbol{S}_i, \boldsymbol{w})$.
11: **end for**

---

from each selected simplex. The complexity of Algorithm 1 is $O(N^{\lceil \frac{K}{2} \rceil} + MKL)$, where $S$ denotes the number of simplices and $L$ denotes the number of simplices.

Based on Algorithm 1, we sample properties $\{\hat{\boldsymbol{c}}^{(i)}\}_{i=1}^M$ and assess generative models in two aspects: 1) General generation quality and diversity. 2) Accuracy to the given mechanical properties.

**General quality and diversity**. We adopt the Fréchet Inception Distance (FID) metric. FID compares the dataset images $\boldsymbol{x}$ with the model-generated images $\tilde{\boldsymbol{x}}$ and is computed as follows:

$$\text{FID}(\tilde{\boldsymbol{x}}, \boldsymbol{x}) = \|\boldsymbol{\mu}_{\tilde{\boldsymbol{x}}} - \boldsymbol{\mu}_{\boldsymbol{x}}\|_2^2 + \text{Tr}(\boldsymbol{\Sigma}_{\tilde{\boldsymbol{x}}} + \boldsymbol{\Sigma}_{\boldsymbol{x}} - 2\sqrt{\boldsymbol{\Sigma}_{\tilde{\boldsymbol{x}}}\boldsymbol{\Sigma}_{\boldsymbol{x}}}) \tag{8}$$

**Mechanical property accuracy.** We pre-train a ResNet18 He et al. (2016) network $\mathcal{F}$ to predict mechanical properties of microstructure images. For a given threshold $\gamma \in [0, 1]$, a generated image $\hat{\boldsymbol{x}}$ is considered accurate with respect to the target properties $\boldsymbol{y}$ if and only if, for every property $j$:

$$|\mathcal{F}(\hat{\boldsymbol{x}}) - \boldsymbol{y}|_j \leq \gamma \cdot |\boldsymbol{y}_j|. \tag{9}$$

The corresponding accuracy is then measured as follows:

$$\text{Acc}_\gamma = \frac{1}{M} \sum_{i=1}^M \delta \left[ \bigwedge_j \left| \mathcal{F}(\hat{\boldsymbol{x}}^{(i)}) - \boldsymbol{y}^{(i)} \right|_j \leq \gamma \cdot |\boldsymbol{y}_j^{(i)}| \right]. \tag{10}$$

where $\gamma \in \Gamma \in \{0.1, 0.2, 0.3\}$. The average accuracy across different $\text{Acc}_\gamma$ values as follows:

$$\text{mAcc} = \frac{1}{3} \sum_{\gamma \in \Gamma} \text{Acc}_\tau. \tag{11}$$

Furthermore, to ensure the trained evaluation network focuses on structural and textural features rather than global luminance, we apply random gamma correction and contrast stretching to all images before feeding them into the evaluation network during training. Specifically, let $\mathbf{x}$ denote the original image, which is normalized to $[-1, 1]$. We perform the following transformation:

$$\tilde{\mathbf{x}} = \text{Clip} \left( \alpha \cdot \left( \frac{\mathbf{x} - \min(\mathbf{x})}{\max(\mathbf{x}) - \min(\mathbf{x})} \right)^\eta \right), \tag{12}$$

where $\eta \sim \mathcal{N}(1, \sigma_\eta^2)$ is a randomly sampled gamma parameter, $\alpha$ is a contrast stretching coefficient, and $\text{clip}(\cdot)$ denotes clipping to the valid range. This preprocessing suppresses low-frequency (luminance) information and enhances high-frequency (texture) details, making the evaluation more sensitive to structural fidelity and less affected by brightness variations. We empirically find that this normalization improves the fairness and robustness of the accuracy metric.

Lastly, the appendix explains the idea of Algorithm 1 more clearly, and the appendix presents experimental results of samples obtained from naive sampling in the whole space.

# 4 EXPERIMENTS

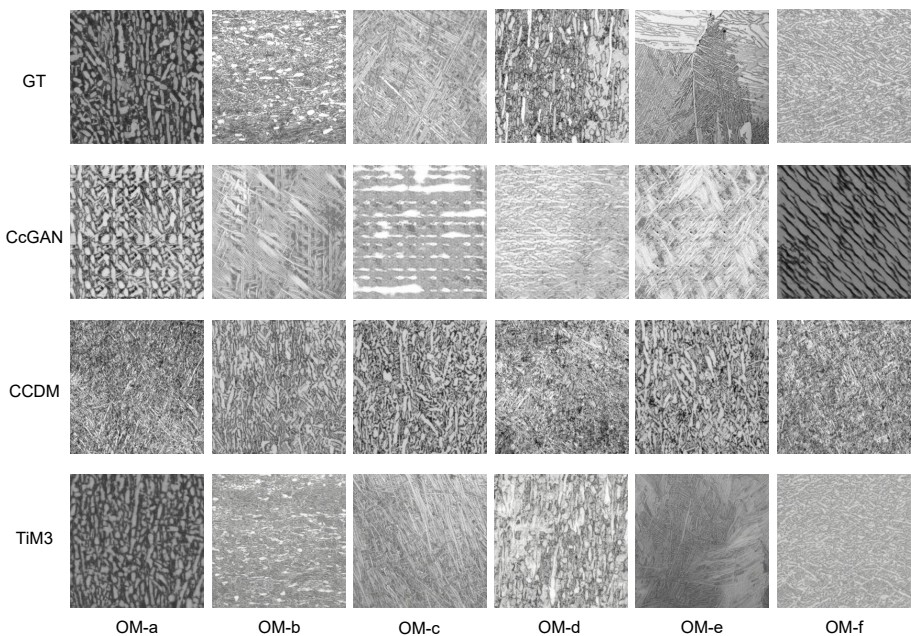

Figure 4: Qualitative comparison on the OM dataset. The "GT" refers to the ground truth image associated with the corresponding mechanical property vector. TiM3 demonstrates exceptional microstructure quality, diversity, and accuracy in comparison.

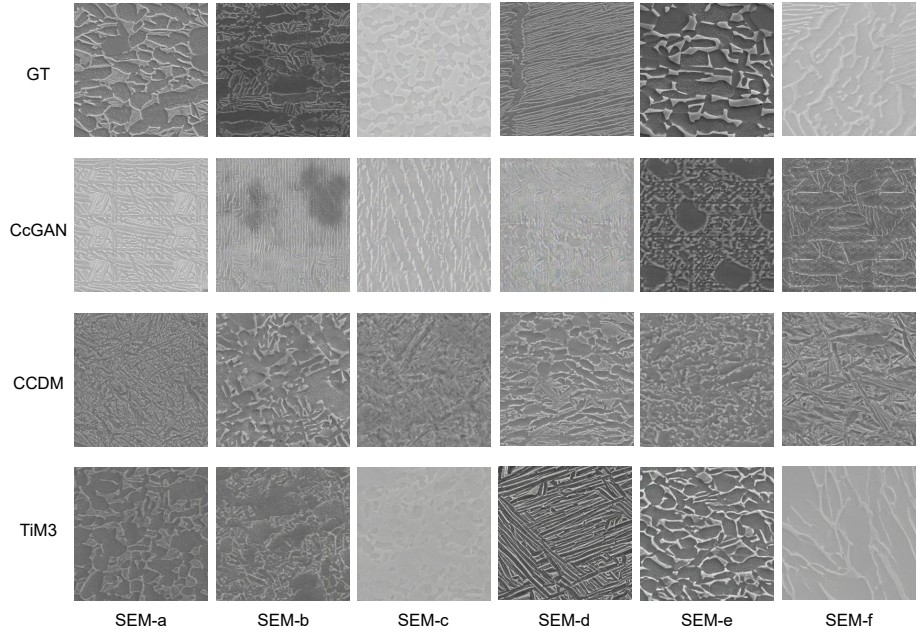

Figure 5: Qualitative comparison on the SEM dataset. The "GT" refers to the ground truth image associated with the corresponding mechanical property vector. TiM3 exhibits outstanding microstructure quality, diversity, and accuracy in this comparison.

## 4.1 Experimental Setup

**Dataset**. We collected paired samples for Ti555 armor titanium alloys, including both optical microscope (OM) and scanning electron microscope (SEM) images along with their corresponding mechanical properties. The OM dataset comprises 79 property vectors, while the SEM dataset includes 123 property vectors, with a total of 124 distinct property vectors. Each vector encompasses 7 mechanical properties: "Yield Strength (YS)", "Tensile Strength (TS)", "Elongation after Fracture (EF)", "Rockwell Hardness (RH)", "Impact Toughness (IT)," "Dynamic Compression Strength (DCS)", and "Critical Fracture Strain (CFS)". The raw microstructure images of the alloys are large, so we divided them into $256 \times 256$ smaller regions. After processing, the OM dataset contains 28,587 sample pairs, and the SEM dataset has 21,872 sample pairs for training.

**Implementation Details**. We train two separate models on the OM and SEM datasets for 50 epochs with a batch size of 32. All experiments are conducted on a single NVIDIA A6000 GPU (48GB). The downsampling rate $f$ of the image encoder is set to 8, and the default latent space shape was set to $32 \times 32$ with a channel size of 4, resulting in generated images of $256 \times 256$. We added zero-mean Gaussian noise to the input mechanical properties. The range of mechanical properties and noise parameters is detailed in the appendix. During inference, we used DDIM Song et al. (2020) with the number of steps set to 20 for efficient generation. To evaluate the models' performance, we sampled 10,000 mechanical property vectors using Algorithm 1.

## 4.2 Main Results

The main experimental results are presented in Table 1. For comparison, we extended CcGAN Ding et al. (2021) and CCDM Ding et al. (2024) to handle multiple continuous conditions, and further details are provided in the appendix.

**Outstanding Mechanical Property Accuracy**. On both OM and SEM datasets, TiM3 achieves state-of-the-art (SOTA) property accuracy, significantly outperforming CcGAN and CCDM. Specifically, on the OM dataset, CCDM shows better accuracy than CcGAN, but TiM3 attains a mAcc of $50.4\%$, surpassing CCDM by $21.8\%$. On the SEM dataset, CcGAN performs better than CCDM, yet TiM3 achieves a mAcc of $44.4\%$, outperforming CcGAN by $26.0\%$. Moreover, TiM3's superior accuracy is consistent across different error tolerances. For all thresholds $\tau \in 0.1, 0.2, 0.3$, TiM3 demonstrates significantly better performance than both CcGAN and CCDM. These results confirm the high accuracy of TiM3 in matching generated images with specified mechanical properties.

**Outstanding Image Quality and Diversity**. TiM3 achieves superior FID scores compared to the baseline methods. Specifically, TiM3 surpasses the second-best method, CcGAN, by 48.3 on OM image generation and by 11.1 on SEM image generation. Meanwhile, CCDM shows notably inferior performance, with FID scores significantly higher than TiM3 on both datasets. These results demonstrate TiM3's outstanding advantage in overall image quality and diversity. The superior performance of TiM3 can be attributed to its effective modeling of texture-level features, which are crucial for high-quality image generation in microscopy domains.

Table 1: TiM3 achieves state-of-the-art accuracy compared to baseline methods. The unit of "latency" is in seconds.

| Method | OM | | | | | SEM | | | | | latency |
|--------|-----|------|--------------|--------------|--------------|------|------|--------------|--------------|--------------|---------|
| | FID | mAcc | $\text{Acc}_{0.1}$ | $\text{Acc}_{0.2}$ | $\text{Acc}_{0.3}$ | FID | mAcc | $\text{Acc}_{0.1}$ | $\text{Acc}_{0.2}$ | $\text{Acc}_{0.3}$ | |
| CcGAN | 93.6 | 12.3 | 0.2 | 4.15 | 32.5 | 82.3 | 13.2 | 0.3 | 8.7 | 30.6 | **0.02** |
| CCDM | 123.6 | 28.6 | 1.2 | 22.9 | 61.7 | 54.4 | 18.4 | 0.6 | 13.4 | 41.2 | 1.33 |
| TiM3 | **45.3** | **50.4** | **2.8** | **60.3** | **88.2** | **43.3** | **44.4** | **1.8** | **48.3** | **83.2** | 2.98 |
| $\Delta$ | ↓48.3 | ↑21.8 | ↑1.6 | ↑37.4 | ↑26.5 | ↓11.1 | ↑26.0 | ↑1.2 | ↑34.9 | ↑42.0 | - |

## 4.3 Ablation Study

To evaluate the effectiveness of core components of TiM3, we also perform ablation experiments, as presented in Table 2.

**Probabilistic Property Encoder**. Compared to the baseline method, incorporating the probabilistic encoder shows a significant improvement in property accuracy. Specifically, on the OM dataset, the mAcc increases substantially from 35.8 to 49.1. And on the SEM dataset, mAcc improves from 41.5 to 42.5. However, the improvement of property accuracy sacrifices general image quality, as the FID increases from 83.0 to 93.4 on the OM dataset and from 42.6 to 53.5 on the SEM dataset. These results indicate that the probabilistic encoder successfully enhances the precision of property control, albeit with a compromise in general image quality.

**Two-Stage Generation** Adopting the metallography-focused two-stage generation pipeline achieves outstanding performance. Specifically, for general image quality, the final model achieves an FID of 45.3 on the OM dataset, which outperforms the baseline by 37.7. And on the SEM dataset, the FID is comparable to the baseline. More importantly, the two-stage generation pipeline preserves higher property accuracy, where the mAcc increases from 35.8 to 50.4 on the OM dataset and from 41.5 to 44.4 on the SEM dataset. These results demonstrate our core contribution: **maintaining generation quality and diversity while significantly improving property accuracy**.

Table 2: Ablation study. "Baseline" is a latent diffusion model with a multi-layer perception property encoder. Both probabilistic encoder and two-stage generation are proven to be effective.

| Method | OM | | | | | SEM | | | | |
|--------|-----|------|-------------|-------------|-------------|-----|------|-------------|-------------|-------------|
| | FID | mAcc | $Acc_{0.1}$ | $Acc_{0.2}$ | $Acc_{0.3}$ | FID | mAcc | $Acc_{0.1}$ | $Acc_{0.2}$ | $Acc_{0.3}$ |
| Baseline | 83.0 | 35.8 | **2.8** | 35.3 | 69.2 | **42.6** | 41.5 | 0.3 | 42.9 | 81.4 |
| + Probablistic Encoder | 93.4 | 49.1 | 2.6 | 58.5 | 86.4 | 53.5 | 42.5 | 0.6 | 45.4 | 81.5 |
| + Two-stage Generation | **45.3** | **50.4** | **2.8** | **60.3** | **88.2** | 43.3 | **44.4** | **1.8** | **48.3** | **83.2** |

## 4.4 QUALITATIVE COMPARISON

We generate images using different methods for qualitative comparison. Six mechanical property vectors were selected from the training set for both OM (Figure 4) and SEM (Figure 5) datasets. The "GT" row includes the ground truth images corresponding to these property vectors.

**General Quality and Diversity**. As illustrated in Figure 4-5, TiM3 is capable of generating images with distinct structures, such as fibers, nets, ellipses, and more. Additionally, generated images exhibit high diversity, covering various microstructure types of titanium alloys, including equiaxed (OM-a), bimodal (OM-b), basketweave (OM-c), and lamellar (SEM-d). In contrast, CCDM tends to produce more homogeneous microstructures with lower diversity, while CcGAN struggles with artifacts (SEM-b) and unrealistic spatial periodicity (SEM-a). These qualitative results demonstrate the superior image quality and diversity of TiM3.

**Accuracy**. TiM3 generates microstructure images that are closer to the ground truth structures in Figure 4 and Figure 5 compared to CcGAN and CCDM. Not being explicitly conditioned on structure types, TiM3 accurately replicates the same types as the ground truth structures, which significantly influence the mechanical properties of the titanium alloy. These results further explain TiM3's high accuracy in generating images that correspond to the given mechanical properties.

## 5 CONCLUSION

In this paper, we propose TiM3 to address the microstructure generation challenge for titanium alloys, guided by multiple mechanical properties, thereby facilitating material design. Since the mechanical properties are continuous conditions, TiM3 employs a probabilistic property encoder to enhance the model's robustness. By adopting the metallography as the intermediate representation, we propose a two-stage generation pipeline. For evaluation, we introduce a method to sample property vectors within the convex hull of the dataset to balance generalizability and the authenticity of property vectors. TiM3 achieves state-of-the-art accuracy in generating both OM and SEM images, demonstrating its effectiveness. Despite the promising results, our work focuses on a single type of titanium alloy; extending support to more types could be a valuable avenue for future work.

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

## A  DETAILS OF CCDM AND CCGAN

CcGAN Ding et al. (2021) and CCDM Ding et al. (2024) are two continuous condition generative models that perturb the input condition with strength $\kappa$ and adopt kernel estimation for condition smoothing. Both methods are based on single continuous labels, which cannot be directly adopted for multiple mechanical properties. To extend them to our task, we make two major changes. Firstly, we adopt $\kappa$ in each dimension of given property vectors; the values of $\kappa$ are computed in the same way as they are calculated for single conditions. Secondly, considering the category distribution characteristics of our titanium alloy dataset, we set the upper limit for the number of images under each property vector to 64, to alleviate category imbalance. More details of training setups can be viewed in Table 3.

Table 3: Training setups of CcGAN and CCDM.

| Dataset | Method | Training Setup |
|---|---|---|
| OM | CcGAN | steps=50K, hard vicinity, generator batch size=64, discriminator batch size=64, $\sigma_{mean}$=0.0228, $\kappa_{mean}$=0.1708, generator lr=1e-4, discriminator lr=1e-4, Adam($\beta_1$=0.5, $\beta_2$=0.999) |
| | CCDM | steps=50K, hard vicinity, batch size=24, $\sigma_{mean}$= 0.0228, $\kappa_{mean}$= 0.1708, timesteps in training = 1000, timesteps in sampling = 20, lr=1e-4, Adam($\beta_1$=0.9, $\beta_2$=0.99) |
| SEM | CcGAN | steps=50K, hard vicinity, generator batch size=64, discriminator batch size=64, $\sigma_{mean}$=0.0228, $\kappa_{mean}$=0.1708, generator lr=1e-4, discriminator lr=1e-4, Adam($\beta_1$=0.5, $\beta_2$=0.999) |
| | CCDM | steps=50K, hard vicinity, batch size=24, $\sigma_{mean}$= 0.0228, $\kappa_{mean}$= 0.1708, timesteps in training = 1000, timesteps in sampling = 20, lr=1e-4, Adam($\beta_1$=0.9, $\beta_2$=0.99) |

## B  AUGMENTATION OF MECHANICAL PROPERTIES

To further raise the model's robustness to unseen continuous mechanical properties, when training the conditional diffusion model, we add zero-mean Gaussian noise to input conditions, and the noise standards for each property are presented in Table 4.

Table 4: The range of mechanical properties and the strength of the random noise (i.e., "Noise Std.").

| | YS | TS | EF | RH | IT | DCS | CFS |
|---|---|---|---|---|---|---|---|
| Min Value | 800 | 500 | 1.5 | 25 | 4 | 1200 | 0.04 |
| Max Value | 1400 | 1200 | 20 | 60 | 60 | 1800 | 0.3 |
| Noise Std. | 50 | 50 | 1.5 | 3 | 2 | 70 | 0.02 |

## C  MORE EXPLANATION OF ALGORITHM

In reality, different mechanical properties might have complex constraints on each other. For example, the tensile strength is commonly negatively correlated with the impact toughness and positively correlated with the Rockwell hardness. We assume those constraints correspond to convex domains as illustrated in Figure 6 (a) and (b). The intersection of such domains will still be convex as shown in Figure 6 (c). Since the property vectors in the dataset are authentic, the convex hull of the dataset should lie in the authentic domain as illustrated in Figure 6 (d). For this reason, we design Algorithm 1 for sampling evaluation of mechanical property vectors, to balance testing models' generalizability and property authenticity. Moreover, we also try naive sampling in the whole high-dimensional space, and the results are presented in Table 5. TiM3 shows SOTA performances under

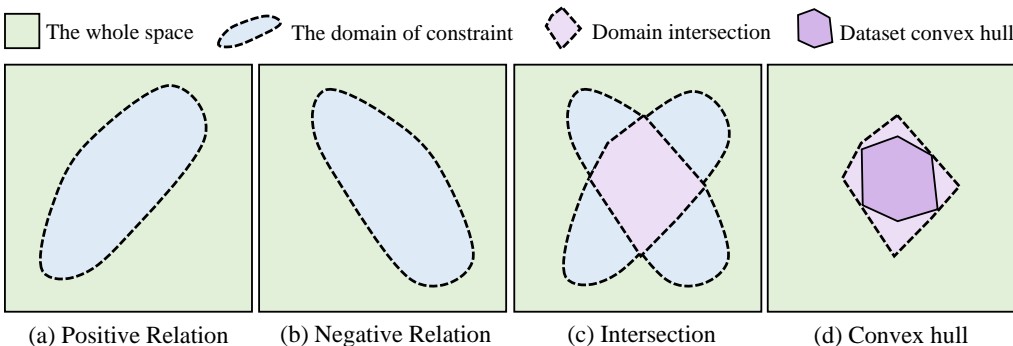

| (a) Positive Relation | (b) Negative Relation | (c) Intersection | (d) Convex hull |

Figure 6: Intuition of adopting Algorithm 1.

Table 5: Quantitative results under evaluation properties obtained from different sampling methods.

| Method | Alg. 1 | OM | | SEM | |
|---|---|---|---|---|---|
| | | FID | mAcc | FID | mAcc |
| CcGAN | | 76.2 | 0.4 | 51.8 | 0.6 |
| CCDM | | 122.1 | 0.9 | 82.3 | 0.7 |
| TiM3 | | **42.8** | **1.92** | **26.0** | **2.84** |
| CcGAN | ✓ | 93.6 | 12.3 | 54.4 | 18.4 |
| CCDM | ✓ | 123.6 | 28.6 | 82.3 | 13.2 |
| TiM3 | ✓ | **45.3** | **50.4** | **43.1** | **44.4** |

the naive sampled properties, indicating the high generalizability of TiM3 even given that properties might not be authentic.

## D  METALLOGRAPHIC IMAGE SEGMENTATION MODEL

In order to obtain metallographic images of titanium alloys that distinguish different phases, we pretrain a segmentation model combining U-Net and attention to segment different phases in metallographic gray images. We fine-tune the U-net framework and add multiple attention mechanisms to enhance the recognition ability of the model for color, texture, and other metallographic features. The original metallographic images and marked metallographic images provided by titanium alloy professional researchers were input into the model for training. The dataset for segmentation model training includes 70 images under OM and SEM and the corresponding complete annotation data.

## E  BRIGHTNESS-CONTROLLED GENERATION

In addition to the seven original mechanical property conditions, we further investigate the capability of our model to control image generation through brightness information. We extend our conditioning framework by incorporating the 8th condition that explicitly represents the target brightness level of the generated image, which is normalized to $[-1, 1]$.

To evaluate the effectiveness of brightness control, we conduct experiments by systematically varying the brightness condition from $-1$ to $1$ with a step size of $0.1$, resulting in 21 different brightness levels. For each brightness level, 1,000 images are generated, and the same mechanical properties are adopted. We then measure the correlation between the target brightness condition and the actual brightness of generated images, which is shown in Table 6.

The experimental results demonstrate a strong correlation between the brightness condition and the generated image brightness across most conditioning settings. As shown in Figure 7, our model successfully generates images with brightness levels that closely match the specified targets. The average absolute difference between target and generated brightness is $0.168$ for OM images and $0.140$ for SEM images, indicating effective brightness control.

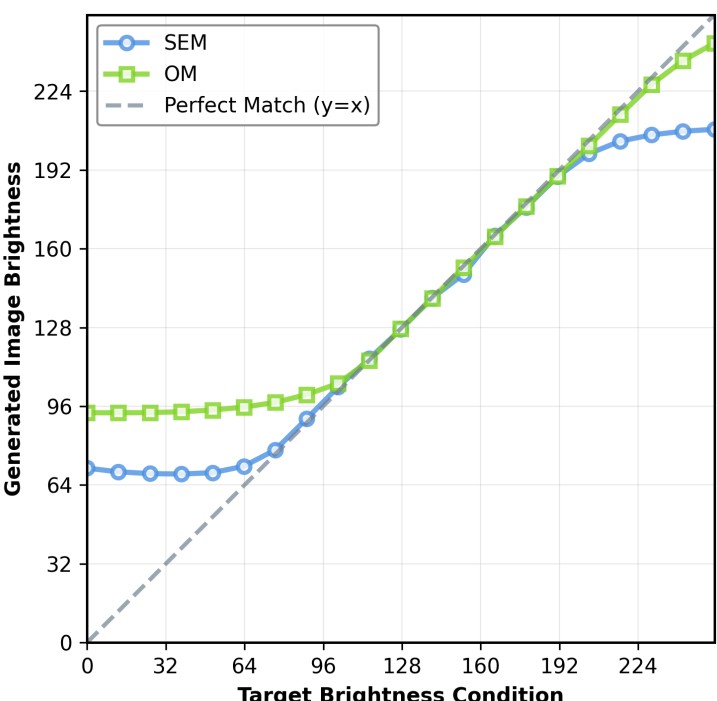

Figure 7: Brightness generation quality.

Table 6: Experimental results of brightness conditioned generation. "B.D." means brightness difference.

|  | FID | mAcc | $Acc_{0.1}$ | $Acc_{0.2}$ | $Acc_{0.3}$ | B. D. |
|---|---|---|---|---|---|---|
| SEM | 50.7 | 35.7 | 2.2 | 31.9 | 73.1 | 0.140 |
| OM | 92.5 | 35.0 | 2.1 | 34.4 | 68.5 | 0.168 |

## F  STAGE 2 GENERATION

To better illustrate the relationship between metallographic images and microstructure images in stage 2 of TiM3, we provide some cases in Figure 8. In the semantic maps of Figure 8, red regions represent $\alpha$ phase, cyan regions denote $\beta$ phase, and yellow regions indicate secondary $\alpha$ structures.

## G  EFFECT OF DIFFERENT DDIM STEPS

In order to explore the generation effectiveness of different DDIM generation steps, we select steps from $\{1, 5, 10, 15, 20\}$ for comparison. The results show that when the number of steps is smaller, the property accuracy is lower.

Table 7: Experimental results with different DDIM steps.

| dataset \ steps | 20 | 15 | 10 | 5 | 1 |
|---|---|---|---|---|---|
| OM | 50.4 | 30.2 | 27.4 | 21.5 | 18.6 |
| SEM | 44.4 | 32.1 | 29.3 | 23.7 | 19.5 |

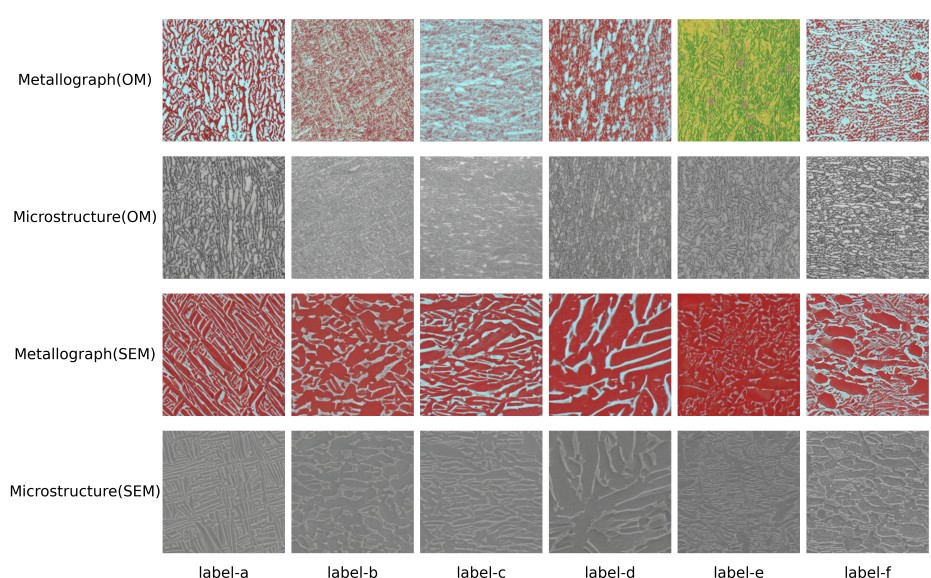

Metallograph(OM)

Microstructure(OM)

Metallograph(SEM)

Microstructure(SEM)

label-a    label-b    label-c    label-d    label-e    label-f

Figure 8: Microstructure Image Generation Based on Metallographic Images.

