# OpenReview forum: "TiM3: Titanium Alloy Microstructure Generation Following Multiple Mechanical Properties"
_ICLR.cc/2026/Conference — Submitted to ICLR 2026_

### Official Review · Reviewer_6WoY · 2025-10-30

**Soundness:** 4
**Presentation:** 3
**Contribution:** 3
**Rating:** 8
**Confidence:** 3

**Summary:**

Existing microstructure image generation methods mainly focus on unconditional or discrete conditional applications, failing to support multiple continuous mechanical properties, in this work the authors suggest a new method to generate images that ensure desired mechanical properties. The work is based on two previous contributions: LDM Rombach et al 2022 and DDIM Song et al 2020. Their main improvement is a probabilistic embedding method based on the Gaussian mixture model to capture continuous mechanical properties. The experiments  are conducted on the data sets the the authors collected, and a computational comparison to state of art is provided.

**Strengths:**

Continuous mechanical property embedding is novel improvement. The metrics of evaluation of mechanical properties are also good contributions. While the work is based on previous techniques, it makes sufficient contributions with convincing experimental results.

**Weaknesses:**

It would be important to report the (time) complexity analysis of this method, particularly with comparision to the state of art.
The authors assume the domain of each constraint to be convex, and then they sample M property vectors uniformly within the convex hull. It would be good to justify this assumption for the convex hull.

**Questions:**

How generalizable is this technique to other material design problems such as soft materials.?

---

### Official Review · Reviewer_JLww · 2025-10-31

**Soundness:** 3
**Presentation:** 3
**Contribution:** 2
**Rating:** 2
**Confidence:** 3

**Summary:**

The paper proposes TiM3, a two-stage latent diffusion framework for generating titanium-alloy microstructure images conditioned on multiple continuous mechanical properties (e.g., yield strength, hardness, toughness). The model introduces a probabilistic property encoder to improve robustness to unseen continuous conditions and uses a metallography-aware two-stage pipeline: first generating a coarse metallographic phase map and then refining it into a realistic micrograph. A convex-hull sampling method is proposed to evaluate generalization in high-dimensional property space. Experiments on Ti555 alloy datasets (optical and SEM images) show lower FID and higher property-following accuracy (mAcc) than adapted CcGAN and CCDM baselines.

**Strengths:**

Clear motivation from materials science. The paper addresses a practical need for controllable microstructure synthesis based on physical property targets.

Sound engineering design. The two-stage “metallography → micrograph” pipeline effectively enforces morphological realism and improves visual plausibility over direct diffusion.

Thoughtful evaluation design. The convex-hull sampling strategy for continuous property evaluation is a well-reasoned attempt to avoid infeasible target combinations.

Readable presentation. Figures and tables are clear; the method is easy to reproduce from the described pipeline.

**Weaknesses:**

- Overclaimed motivation.
The paper states that “existing methods can’t handle multiple continuous mechanical properties,” which is overstated. Conditional generation for continuous and even multi-dimensional labels has been widely explored in prior work (e.g., CcGAN, CCDM, ControlNet-style diffusion, or CLIP-guided conditioning). The true contribution lies in adapting these techniques to titanium microstructures, not in resolving a fundamentally unsolved ML problem.

- Limited methodological novelty.
The probabilistic property encoder is a minor variation on label smoothing or stochastic embedding, and the two-stage structure closely mirrors existing semantic-map-to-image or ControlNet-style diffusion pipelines. The main contribution lies in domain adaptation rather than new generative principles.

- Evaluation fairness concerns.
Appendix A shows that all models were trained for the same 50 K steps with identical optimizers and learning rates on a single A6000 GPU—making the setup superficially fair but not genuinely optimized. The baselines (CcGAN, CCDM) were originally single-label models; extending them to 7-D conditioning without parameter tuning likely underrepresents their potential performance.

- Limited generalization and scientific validation.
Experiments are restricted to one alloy type (Ti-555). Property-matching accuracy is measured using a ResNet-18 predictor trained on the same dataset rather than physical or experimental validation, so the reported “accuracy” may reflect consistency with a surrogate network, not actual mechanical properties.

- Metrics partly circular.
FID, computed from an Inception network trained on natural images, is not meaningful for microscopy. The proposed mAcc depends entirely on the authors’ own property predictor and convex-hull sampling, which favors interpolation near the training set.

- Domain-specific scope.
The work provides clear practical value for materials design but offers limited insights or methodological advances for the broader ML community, making it more suitable for a domain-specific engineering venue.

**Questions:**

- How sensitive is performance to the number of Gaussian components in the probabilistic encoder?

- Could the authors test generalization on unseen alloy systems or synthetic out-of-hull property vectors to demonstrate robustness?

- Please clarify whether the metallographic segmentation model was trained on the same images used for TiM3 training—if so, could that introduce label leakage?

- Consider comparing against a modern conditional diffusion baseline such as ControlNet or T2I-Adapter to show that improvements are not solely due to architecture depth.

---

### Official Review · Reviewer_BeT3 · 2025-11-01

**Soundness:** 2
**Presentation:** 2
**Contribution:** 1
**Rating:** 4
**Confidence:** 3

**Summary:**

It is natural to conduct conditional diffusion models for Microstructure image generation. Following this flow, this work adapts diffusion generation to titanium alloy microstructure image generation by proposing a property-aware condition generation (stage 1) and conditional diffusion. The experiments show competitive results compared to the other two baselines.

**Strengths:**

1. This work is easy to follow. The idea and method have been clearly presented.

2. This work adapts a traditional image generation framework to domain-specific image generation with domain understanding, which benefits the specific task.

**Weaknesses:**

1. Framework Novelty. Although this work introduces a two-stage diffusion framework, this work is a simple adaptation of existing works. The fine-to-grain diffusion and multi-stage diffusion are generally used in previous works.
2. Experiments are not enough to demonstrate the superiority of this work. It only compares with two methods. More image generative baselines should be included and compared.
3. The ablation study is not well disentangled. More experiments, such as parameter sensitivity, are required to demonstrate the effectiveness.
4. There are some typos and grammatical errors. The work requires further proofreading for publication.

**Questions:**

N/A.

---

### Official Review · Reviewer_116Q · 2025-11-01

**Soundness:** 2
**Presentation:** 2
**Contribution:** 1
**Rating:** 2
**Confidence:** 4

**Summary:**

This paper presents TiM3, a two-stage latent diffusion model that generates titanium alloy microstructure images based on multiple continuous mechanical properties. It uses a probabilistic property encoder for robustness and a coarse-to-fine pipeline that first generates metallographic maps and then detailed microstructures. A convex-hull sampling method is introduced to evaluate accuracy in high-dimensional property space. Experiments show that TiM3 achieves high image quality, diversity, and accuracy.

**Strengths:**

1. The method proposed can work like the authors claim, i.e., the model takes continuous properties as input and generate an image conditioned on them.
2. This paper provides a unique dataset for titanium alloys. If this dataset can be published, it will contribute to the community of titanium alloy study.

**Weaknesses:**

1. The motivation of adopting a two-stage generation pipeline is not clear. As illustrated in Figure 2, the first stage of the pipeline can already generate a high quality "metallography" which shares very similar geometry to the final output, and has richer information with respect to the color. If the first stage can already achieve this, the value of the second stage is questionable.
2. A large portion of the methodology is used to describe a mechanical property sampling algorithm. However, the motivation for this is unclear, and against both conventional machine learning setting and practical needs. For machine learning convention, the dataset should be split into training set and testing set, so that there is no need to sample the property at all. For practical needs, the property should come from the need of practical applications, i.e., being also known. I fail to understand why such sampling matters.
3. The metrics used in the experiments are not convincing. The authors use a machine learning model (ResNet18) to measure the accuracy of generated samples. However, as mentioned above, they use sampled properties as properties, which means both the images and properties are not seen by the ResNet18 model during training and may even be far from the training set owing to the sampling process. Therefore, the reliability of such measurement of accuracy is highly questionable.
4. The dataset scale is very limited. Although the two datasets used are claim to have over 20,000 samples for each of them, the samples are actually cropped segments from some large images (as the authors mention, 123 unique vectors). Such operation makes the data similar to the effect of augmentation of 123 samples, because the segments from one large image can share very similar geometry features. The effect of limited dataset is that the diffusion model may easily remember the feature of each sample and tend to reproduce them.
5. For the qualitative comparison shown in Figure 4 and 5, the authors mention that "six mechanical property vectors were selected from the training set". Therefore, these conditions and images are already known by the diffusion model during training, and it is trivial for the model to reproduce them. As a result, such "qualitative comparison" is not convincing. On the contrary, the authors haven't provided generated images based on unknown properties.
6. The setting for the ablation study is not clear, and the ablation study is not convincing enough. The authors do not specifically mention what is the baseline method, so the comparison is hard to understand. Two key issues are not solved by the ablation study. First, what's the performance for a one stage generation pipeline, keep everything else the same. Second, what's the performance if the mechanical property encoder is not used but the property is directly input into the diffusion model (after a linear layer if needed). Without these issues being solved, the contributions of the paper is not verified.

**Questions:**

None.

---

### Meta-Review · Area_Chair_mU6A · 2026-01-07

**Summary:**

This paper presents a diffusion model for generating titanium alloy microstructure images.
The submission received mostly negative reviews.
The reviewers mainly recognize the engineering design and domain value of the application.
The main concerns from the reviewers were the limited novelty (BeT3, JLww), justification of method design (116Q, JLww), limited evaluation (116Q, BeT3, JLww), and lack of generalization to other alloy types (JLww).
The authors did not provide a rebuttal.
The AC agrees with the reviewers that the weaknesses and critical and do not justify for sufficient contributions.
As such, the AC recommends rejection.

**Reviewer Concerns:**

All reviewers' concerns remain outstanding.

**Reviewer Scores:**

I think all reviewers would keep their original ratings.

---

### Decision · Program_Chairs · 2026-01-26

Reject